# Going Wild in the City—Animal Feralization and Its Impacts on Biodiversity in Urban Environments

**DOI:** 10.3390/ani13040747

**Published:** 2023-02-19

**Authors:** Thomas Göttert, Gad Perry

**Affiliations:** 1Research Center [Sustainability–Transformation–Transfer], Eberswalde University for Sustainable Development, 16225 Eberswalde, Germany; 2Department of Natural Resource Management, Texas Tech University, Lubbock, TX 79409, USA

**Keywords:** Anthropocene, domestication, feralization, urbanization, invasive species, biodiversity in novel ecosystems

## Abstract

**Simple Summary:**

Understanding the impact of urbanization on biodiversity is a crucial task of our time. Here, we reflect on the importance of feralization in the relationship between ongoing urbanization and the worsening biodiversity crisis. Feralization is often viewed as the exact opposite of a domestication process—a perception that we argue is too simplistic. The interrelations between domestication, feralization, and the adaptation of taxa to novel, human-made environments such as cities are complex. Given their unique traits, feral(izing) taxa can play key roles in sustainability, sometimes problematic (i.e., invasive species) but at other times, improving human well-being in urban settings.

**Abstract:**

Domestication describes a range of changes to wild species as they are increasingly brought under human selection and husbandry. Feralization is the process whereby a species leaves the human sphere and undergoes increasing natural selection in a wild context, which may or may not be geographically adjacent to where the originator wild species evolved prior to domestication. Distinguishing between domestic, feral, and wild species can be difficult, since some populations of so-called “wild species” are at least partly descended from domesticated “populations” (e.g., junglefowl, European wild sheep) and because transitions in both directions are gradual rather than abrupt. In urban settings, prior selection for coexistence with humans provides particular benefit for a domestic organism that undergoes feralization. One risk is that such taxa can become invasive not just at the site of release/escape but far away. As humanity becomes increasingly urban and pristine environments rapidly diminish, we believe that feralized populations also hold conservation value.

## 1. Introduction

Humans continue to modify Earth in profound ways, some such as global climate change relatively recently and others as part of long-lasting processes. Here, we focus on the interaction between two of the latter: domestication of other species and the feralization that occasionally follows, and urbanization. Although historically and mechanistically different, we argue that the two interact in important ways. We briefly review each process by itself, then assess the ways they interact in a conservation context.

## 2. Domestication and Feralization

In *domestication*, a wild species of plant or animal is removed from its ecological and evolutionary context and progressively incorporated into the human sphere [1]. During this process, it is subject to strong evolutionary pressures under anthropogenic selection and becomes an increasingly better fit to the human template. Artificial selection during this transformation produces new forms, as when a wolf (*Canis lupus*) is domesticated. Despite what can be striking morphological and genetic changes, the result, in this case the dog (*Canis lupus familiaris* or, under the Kiel school convention, *Canis lupus* f. familiaris), is not generally considered a new species (Figure 1, top) [1]. Young (1985) [2] credited Zeuener (1963) [3] with developing a five-step typology for animal domestication (we note that a similar process can occur in plants, but in this paper focus almost exclusively on animals). The process starts with loose contacts and ongoing wild reproduction (step 1); continuing to confinement in human environments (step 2); initial selective breeding (step 3); planned development of breeds (step 4); and finally, at least in some cases, persecution or even extermination of wild forms (step 5). Early stages of the domestication process may or may not involve human intent. In the case of dog domestication, many authors have speculated that the process started with wolf populations that began following hunter-gatherers in the Pleistocene [4]. Some dog breeds, such as the Canaan dog of Israel, remain semi-wild (at least until recently) [5]; whereas others, such as the Pekingese, are fully dependent on people and human habitation [6]. Wild wolves were subsequently hunted to extermination or near-extermination in many areas. 

During domestication, natural selection is increasingly relaxed as species are progressively moved into human-controlled existence. As a result, normal ecological rules no longer apply. For example, masses in six wolf populations studied by Mech and Paul (2008) ranged from 26.3 ± 0.56 to 35.9 ± 0.45 kg (mean ± standard deviation, with the smallest values representing females only and the largest, males) [7]. In contrast, adult mass among dog breeds varies two orders of magnitude [8]. Similarly, the realized productivity of agriculturally important plants and animals is much higher than could exist under normal wild conditions because the required inputs of energy and nutrients could not be sustained and the loads on female survival would be prohibitive. Although domestication can result in survival of more phenotypic diversity [9], the intense inbreeding often involved can create genetically depauperate forms with little ability to survive in the absence of consistent human upkeep. Chosen varieties are further spread by humans to regions where they would never have arrived, or survived, under natural conditions [4,10].

The term “domestication” is commonly used to describe extreme and long-standing selective pressures for human purposes, as in the case of the dog. Yet shorter relationships can also lead to strong selection pressures, as in the captive propagation of multiple color morphs of the corn snake (*Elaphe guttata*). These forms are rarely seen in nature [11] and are unlikely to survive under natural selection. Thus, we argue that all organisms found in the pet and ornamental plant trades are along the domestication spectrum, though often fairly early in the process. Finally, we argue that any individual that does well in an urban setting is likely to be the result of an (unintentional) selection process that favors forbearance of human nearness, consistent with associations included in *step 1* of the domestication process [3]. Since domestication effects can become entrenched within a few generations when animals are kept under artificial conditions [12], we employ the broader understanding here, thus viewing domestication as a more common phenomenon than is sometimes portrayed.

The process of domestication can be reversed when a tamed form partially or completely escapes husbandry back into the wild, resulting in *feralization* through processes that are not well understood [13]. Reversing the five-step model of Zeuener (1963), captive individuals go from dependence on humans (step 4 or 3) back to a loose association (step 1) or even completely unassociated existence [3]. Bonacic et al. (2019) divided this process into three stages: in the first, a species is domestic and carefully cared for [14]. However, poor husbandry leads to free-roaming (stage II), which in turn leads to a fully feral population lacking human inputs (stage III) [14]. In the example of dogs, feral forms have appeared in many locations, perhaps most famously the fully wild dingo, *Canis lupus dingo*, in Australia (Figure 1, bottom). At the end of the process, a species formerly kept in artificial environments becomes freely reproducing in a more natural environment [14]. Feral dogs, such as the “boonie” dogs of Guam are found in many locations and may in turn be re-domesticated. Other good examples are species originally brought to Europe as fur animals, such as the raccoon (*Procyon lotor*), raccoon dog (*Nyctereutes procyonoides*), American mink (*Neovison vison*), muskrat (*Ondatra zibethicus*), and coypu (*Myocastor coypus*). These species were kept under artificial conditions for generations and underwent partial domestication before escaping (or being released) and becoming self-sustaining faunal elements in environments where they were non-native, even invasive [15]. 

The habitat a newly-feral organism faces may be novel—canids were not naturally found in Oceania prior to human introduction of dogs some 3300 years ago [16]—or quite similar to the region from which the species was originally domesticated. The resulting natural selection pressures can thus be almost identical, or drastically different, to those experienced during the evolution of the originating wild species. In either case, that selection is being experienced by a domesticated form that is genetically and otherwise distinct from the progenitor organism. For example, silver foxes (*Vulpes vulpes*) experimentally domesticated were not just tamer but also showed changes to seasonal reproductive patterns and even morphology [9]. Moreover, captive and thus feral forms of both plants and animals are often the recipients of panmixia, as individuals from multiple wild sources are brought together to create a pet or ornamental population. Examples are the palm *Trachycarpus fortune* [17] and *Anolis* lizards [18].

Price (1984) [19], referenced in Nichols (1991) [20] argues that domestic Japanese quail (*Corturnix japonica*) that became feral on Hawaiian islands and lost their “domesticity” can be treated as wild Japanese quails after many generations. We do not agree with this opinion, since these quails have probably passed a genetic bottleneck during the domestication process leading to a certain degree of genetic depression. Moreover, they are introduced organisms that compete with native organisms and change the food web of the ecosystem. A similar argumentation chain to that of Price can be seen behind the concept of so-called “de-domestication,” defined as “*a sort of species restoration, a way of getting populations of animals to resemble their wild ancestors not only in appearance but also in terms of behavior*” [21] (p. 3). However, even identical natural selection pressures during feralization are unlikely to result in an exact replica of the ancestral form despite some arguments to the contrary. The organism losing domestic features and forming self-sustaining populations in the wild has previously undergone repeated selection processes that ensure it is not the same entity as the originating wild species. A “rewilded” dog is a new entity, not wolf 2.0.

Since only a very small fraction of species have been fully domesticated—less than 0.5% of extant species of Eutheria, for example [1]—feralization may be imagined to be easy to survey. However, depending on which species are considered to be at least partially domesticated, the number of species that *might* become feral can increase rapidly (Table 1, Table 2 and Table 3). When expanding the concept of feralization to plants (some authors use the term “naturalization” instead of feralization for vegetation [22]), the numerical dimension of potentially involved species expands greatly: more than 5000 species of ornamental plants of American origin were introduced to Europe [23].

## 3. Conservation Impacts of Feralization

The number of feral species, some of them damaging invasives, can be locally large. For example, in a study on the aquatic and riparian mammals of the Tierra del Fuego and Cape Horn region of Argentina and Chile, Anderson et al. (2012) listed 13 native species and 17 that are non-native [24]. Of the latter, six have the word “feral” in their name and most of the others fit the broad definition used here. In places, feral populations of individual species can be very large, often as a result of being subsidized by human activities (e.g., Allan et al., 1995, for Canada geese, *Branta canadensis* [25]), and therefore have outsized impacts. With the exception of species considered to be undesirable such as urban pigeons (*Columba livia*), however, the literature on the process and conservation impacts of feralization (as opposed to invasion) is quite sparse, despite its far-reaching ecological dimension. The conservation impacts of such forms include:**Predation on native species**. Despite an abundance of literature on the effects of native predators, such as wolves, on domesticated species, we have not found any comprehensive reviews of similar impacts by most feral species. North American bullfrogs (*Lithobates catesbeiana*), for example, were raised in captivity in many areas in connection with farming for frog leg meat. Once released or escaped, feral populations quickly became major predators of multiple species [26]. Perhaps the most consequential feral predator, however, is the domestic cat (*Felis silvestris* f. catus) [27].**Competition with native species**. Having been widely introduced from captive stocks, North American bullfrogs also effectively outcompeted many native anurans [26]. Similarly, the presence of feral pigs on Santa Cruz Island (California, USA) led to near extirpation of the native Island fox, *Urocyon littoralis* [28]. As with the previous category, however, we found species-specific reviews—e.g., for alfalfa (*Medicago sativa*) [29] and reviews of non-native impacts, which include feral taxa—e.g., for foxes and cats [30] and water hyacinth (*Eichhornia crassipes*) [31], which also causes eutrophication (Figure 2), but not comprehensive reviews of competition between feral and native species.Feral animals that **intermix with their progenitor taxon**. Examples are introgression between domesticated cats and wildcats (*Felis sylvestris*) in Europe [32,33] and hybridization of European honeybees (*Apis mellifera*) with feralized African forms [34].

Feralization, however, is not necessarily a unidirectional process and can be entangled with domestication, as shown by the spread of domestic cattle in central Europe during the course of the Neolithic revolution (Figure 2).

Domesticated forms of the aurochs (*Bos primigenius* f. taurus) from the Middle East reached central Europe several millennia ago. There they were further domesticated and at the same time came into contact with their progenitor species, the aurochs (*Bos primigenius*). This is the classic conflict of hybridization between wild and feral taxa. Currently, there is a similar situation with the yak, where the wild population (*Bos mutus*) runs the risk of being incorporated into the gene pool of the domesticated form (*Bos mustus* f. grunniens) through hybridization. Besides this hybridization, feralization comes into play: Wild males enter domestic herds and abscond with domestic females not considered desirable by herders [35]. The domestic females feralize and the F1 generation (hybrids) lives in a wild environment but may be re-incorporated into the human sphere. Although these hybrids can be difficult to domesticate, animal herders sometimes consider incorporation of hybrid blood lines desirable [35]. As a consequence, the wild gene pool gets messed up and—given the increasing proportion of domestic taxa in an environment compared with wild ones over time—the wild one may become extinct at a certain point. Feralization being interwoven with domestication also explains why we keep seeing piglets in wild boar (*Sus scrofa*) populations that show the typical pattern of domesticated pigs (Figure 3F). Apparently, parts of the genome of domestic pigs were incorporated into the gene pool of wild boar [36]. Wu and others very recently provided evidence for a loss of wild genotype through domestic chicken (*Gallus gallus* f. domestica) introgression during the Anthropocene [37]. Also, wild sheep living in Europe today, which are perceived by many people as natural fauna elements, are nothing more than feral domestic sheep derived from the species *Ovis orientalis* [38].

4.Feral species that become wild in an area where they **hybridize with less closely related native species**. Examples include the plant genus *Typha* [39] and hybridization of feral red-eared sliders (*Trachemys scripta elegans*) with other freshwater species [40].5.**Disease vectoring** occurs when the presence of a feral taxon allows disease-causing organisms to spread to novel native taxa or increases the rate at which such spread occurs. For example, feral pigs have been shown to vector soil-borne plant pathogens in New Zealand [41]. Feral populations of the frog *Xenopus laevis* in Chile are infected with *Batrachochytrium dendrobatidis*, a fungal pathogen causing amphibian population declines, and can spread the fungus to native species [42].6.Cases of **ecological replacement of an extinct ecological equivalent** are hard to assess for impact, since the extinction preceded the arrival of the feral taxon. For example, the European mink (*Mustela lutreola*) disappeared in many parts of its range before the American mink (*Neovison vison*) was introduced. However, in parts of the range the feral introduction contributed to the disappearance of the native species, and attempts to reintroduce the European mink are also hampered where the American counterpart is present [43]. Somewhat similarly, the wild horse (*Equus ferus*) had almost completely disappeared and attempts to rewild it are partially based on domesticated animals [44]. Similar issues with genetic mixing were seen in markhors (*Capra falconeri*) being bred at zoos for reintroduction, over a third of which were found to carry genes from domestic goats (*Capra aegagrus* f. hircus) [45]. A release of such hybrids might meet with conservation objectives, but could arguably be considered an introduction of a new form. Another interesting example of replacement of long-extinct ecological equivalents is provided by fallow deer (*Dama dama*) and its important role in transforming cultural landscapes in medieval England between the 11th and 16th century AD [46]. The conservation consequences of this intentional rewilding appear to be consistent with management choices or at least public wishes. For example, feral rabbits on Okunoshima island in Japan [47] and feral horses of Australia [48] are perceived as non-natural but nevertheless regarded as worthy of protection.7.Some feral taxa take root in **areas where there is no comparable species**. Examples are feral parakeet (*Psittacula*) populations in different cities in Germany and France, [49] or greater rheas (*Rhea americana*) in northern Germany [50]. Such introduction may have no ecological impact on the kinds discussed above, because the ecological niche they invade was previously unoccupied, but other impacts, such as changes in fruit dispersal on sprouting success, as well as competition for nesting places and food with native urban birds, may arise in the case of the parakeet. Moreover, feral parakeets may pose a threat of spreading pathogens to other birds [49]. Although no negative impacts have so far been reported in terms of greater rheas in Germany, they may pose a threat for ground-nesting species owing to nest-trampling [50].8.Finally, the **contribution of feral taxa to human-wildlife conflict** creates public relations problems eroding support for conservation. For example, feral Canada geese and other waterfowl (Figure 3C,D) have proliferated in urban settings to the point where they create a public nuisance [25]. Although a number of species fall into this category, particularly in urban settings, the most consequential example is the domestic cat. This species, while doubtlessly causing extensive depredation of birds and other species, is also beloved by many [27]. The resulting conflict, most notably between wildlife managers and animal rights advocates, is extremely difficult to resolve and contributes to a divide between pro-nature groups that might otherwise find much to cooperate on [51].

In the following, we provide examples for conservation impacts of feralization (Table 1, Table 2 and Table 3). animals-13-00747-t001_Table 1Table 1Selected examples of feralization events of mammalian species and the ecological implications. Temporal information relates to specific examples.Feral TaxonEcological ImpactsRegion (Example)Start of Feralization (Example)ReferenceLagomorpha



*Oryctolgus cuniculus* f. domesticaNegatively affecting natural vegetationOkunoshima, Japan1970s[47]Rodentia



*Mus musculus* f. domestica-Faray Island1940s[52]Carnivora



*Canis lupus* f. familiarisDespite the small population have an enormous negative impact on native faunaGalapagos Islands1930s[53]*Felis silvestris* f. catusIntermixing with progenitor taxon, predator of various taxaSardinia3000yBP[54]*Neovison vison*Predator of various taxa, in competition with native mustelidsGermany1950s[15]*Mustela putorius* f. furoPredator of various taxaNew Zealand1880s[55]Artiodactyla



*Camelus dromedarius*Negatively affecting natural vegetationAustralia1908–1911[56,57] *Lama guanicoe* f. glamaFailed to become establishedAustralia1900[56]*Dama dama*Transforming cultural landscapeUK11th–16th century[46]*Bos primigenius* f. taurusTransforming cultural landscape, surrogate taxon → “rewilding”Oostvaardersplassen, Netherlands1980s[58]*Bos gaurus* f. frontalisIntermixing with progenitor taxonBangladesh?[59]*Bos mutus* f. grunniensIntermixing with progenitor taxonHelan mountains, China?[35]*Bubalus arnee* f. bubalisNatural vegetation Western Australia1850s[56,60]*Capra aegagrus* f. hircusVarious taxonomic groups affected, severe impact on vegetationGalapagos islands1813[61]*Ovis orientalis* f. aries-Shackleford Banks, island, USA1940s[62]*Sus scrofa* f. domesticaSignificant ecological damage, various taxaSouth AmericaSince 1493[63]Perissodactyla



*Equus ferus* f. caballusOvergrazing, affecting reproductive characteristics of dominant grass speciesAssateague Island, USA1500s[64]*Equus africanus* f. asinusDeclared as “vermin” in some regionsKimberleys, Australia1930s[56]
animals-13-00747-t002_Table 2Table 2Selected examples of feralization events of avian species and the ecological implications.Feral TaxonEcological ImpactsRegion (Example)Start of Feralization (Example)ReferenceGalliformes



*Gallus gallus* f. domesticaPossibly reservoirs and vectors of aviandiseasesSeveral Galapagos islandsLate 1990s[53]*Pavo cristatus*-UK1980s[65]*Numida meleagris*Failed to become establishedAustralia1920s[56]*Corturnix japonica*Important gene pool as native population decreasingHawaiian islands1920s-1950s[20]*Meleagris gallopavo*-Hawke’s Bay, New Zealand1860s[66]Columbiformes



*Columba livia* f. domesticaDeclared as “vermin” in some regionsWestern Australia1950s[56]Anseriformes



*Alopochen aegyptiacus*-The Hague, Netherlands1967[67]*Anser anser*Co-existing with autochthonous individuals of the same speciesGermany1980s[68]*Branta canadensis*Competition with native waterfowlUK1660s[25]*Anas platyrhynchos*Hybridization with native *Anas gracilis*New Zealand1862[69]*Aix galericulata*-Madeiran arc hipelago2010[70]Passeriformes



*Acridotheres tristis*Predation and aggressive competition with native wildlifeCanberra, Australia1968[71]Psittaciformes



*Melopsittacus undulatus*-Florida, USALate 1950s[72]*Psittacula krameri*Potentially becoming a disease vectorGermany, France 1960 and 1970s[49]*Myiopsitta monachus*-New York, USALate 1960s[72]*Pyrrhura molinae*-Texas, USA1990s[72]*Brotogeris versicolurus*-California, USA1970s[72]*Brotogeris chiriri*-California, USA1970s[72]*Amazona viridigenalis*-California, USA 1990s[72]Rheiformes



*Rhea americana*Possibly nest-trampling of ground-nesting bird speciesNorthern Germany2000[50]
animals-13-00747-t003_Table 3Table 3Selected examples of feralization events of other taxa and their ecological implications. Taxon names in quotation marks indicate paraphyla or historically used systematic groups.Feral TaxonEcological ImpactsRegion (Example)Start of Feralization (Example)Reference**“Reptiles****”**



*Trachemys scripta*Competition with native freshwater tortoisesSicily, Italy1990s[73]**Lissamphibians**



*Lithobates catesbeiana*Competition with other amphibians, predator of various taxaSantay Island, Ecuador1990s[74]**“Osteichthyes****”**



*Cyprinus carpio*Because of early introduction difficult to assess Central Europe1st–2nd century[75]*Oncorhynchus mykiss*Threat to localized biota, various taxaItalySince 1890s[76]*Carassius gibelio* f. auratusCompetition with native *Carassius carassius*UK17th century [77]**Insecta**



*Apis mellifera scutellata*Hybridization with European *Apis mellifera*Mexico1980s[78]


## 4. Urbanization

Like domestication, the origins of urbanizations extend thousands of years before historical records began [79]. Early urban efforts were culturally significant but demographically negligible, and residents were the most “*disease-ridden and the shortest-lived populations in human history*”, as Cohen wrote in terms of ancient Europe [80] (p. 127). Today, over half the human population lives in cities and urban life expectancy and quality of life often greatly exceed those in outlying areas [81]. The fraction of humanity living in cities is increasing rapidly, most speedily in Africa and Asia. Throughout the world, however, most people do not live in the large (>5 million inhabitants) mega-cities that tend to capture media attention. Instead, smaller municipalities, typically under half a million residents and often referred to as secondary cities, are humanity’s primary dwelling places [82].

Cities of all sizes do not just house people and the infrastructure that supports their economic activities. Urban green spaces (UGS) have beneficial impacts on human well-being [83,84], economic development [85], and more. Many other species find habitat in urban spaces, often but not always in UGS [86]. UGS can be found in cities around the globe, though access is not uniform and is confounded by issues of social justice [87,88].

## 5. Conservation Impacts of Urbanization

Urbanization involves large-scale, intensive, and typically irrevocable land use conversion. As this transformation progresses, initially small changes [89] become increasingly larger and more catastrophic for many native species [90]. Much has been made of the importance of land conversion for urbanization and the resulting loss of biodiversity [91,92], especially direct impacts in high-income country contexts [90]. Here, we focus on the *ongoing* presence of wildlife, some of it native, within urban settings and its implications for conservation: we are interested in the urban environment as a novel ecosystem *sensu,* as presented in Hobbs et al. [93].

Although much of the assessment of urban wildlife has been in the setting of human-wildlife conflict [84,86,91], awareness of non-conflict urban wildlife also goes back decades [94,95]. Concerns about the conservation value of urban settings often emphasize that, as anthropogenically modified habitats, they are particularly amenable to non-native, globally distributed species such as urban pigeons [96] and taxa taking advantage of the ample night-light niche [97]. Another common concern is that some of the animal species commonly associated with humans are often kept as pets in urban settings, where they can get loose and perhaps survive to become at least somewhat feral. 

On the positive side, cities provide habitat for the subset of species that can coexist with humans, some of them tolerant natives and others synanthropic [90]. Although some of those are typically seen as pests, others are less common or even of conservation concern. For example, the red fox (*Vulpes vulpes*) has successfully colonized multiple European cities, including Zurich, Switzerland [98] and Berlin, Germany [99]. Similarly, Sosa (2009) studied urban ornate box turtles (*Terrapene ornata ornata*) in Texas, native members of a genus that is declining and of substantial concern throughout its range, and found persistent populations in yards and UGS [100]. Finally, several raptor species do very well in urban settings, some of them highly threatened in other settings [101].

## 6. Domestication, Feralization, and Urbanization: A Conservation Perspective

Cities are the quintessential “novel” or “anthropogenic” ecosystems—not just human-dominated, as an agricultural area is, but artificial from below the surface to the tops of sometimes sky-scraping buildings. Applying the term “ecosystem” to places like cities runs afoul of the traditional view of ecosystems as self-organizing if inconstant entities [102]. Novel ecosystems like cities rely upon ongoing human inputs, and in that sense explicitly include humanity as an ecosystem component, a somewhat controversial notion. Nonetheless, some species are able to take advantage of the novel resources provided by persistent human presence [103]. We consider the concerns of Murcia et al. [102], Simberloff [104], and others very valid and do not believe they have been fully resolved. Nonetheless, we also believe that the potential of UGS and other urban spaces to provide some conservation benefits should be explored, while taking into full consideration the negative impacts of the ongoing urbanization process.

Traditionally, views of domestication do not consider it a process that happens in an urban setting. Rather, ancient societies are often visualized as domesticating wild species for their usefulness, often in an agricultural setting. Modern cities are not friendly to most of these, though settlements in the Global South often house livestock for various uses (Figure 4A–D) [105]. A resident of a major Global North city, in contrast, might need to go to a zoo to see a sheep or a goat (Figure 4E). And yet, cities are hotbeds of ongoing domestication because they house both fully tame organisms (cats, dogs, some ornamental plants) and numerous species in earlier stages of domestication, originating in the pet and ornamental plant trades. Some of these are wild-caught or dug up in nature, but many are the result of generations of human cultivation, sometimes with strong artificial selection for particularly desirable phenotypes. Domesticated taxa are often thought to become unable to survive outside of human cultivation, and that certainly occurs in some cases [106]. For example, there are several species that have been domesticated and are ubiquitous all over the world including cities, but which, to our knowledge, have not yet become feral anywhere. This applies to the various domesticated hamster species, the domesticated guinea pig (*Cavia aperea* f. porcellus) or the domesticated canary (*Serinus canaria* f. domestica). But other species do succeed in feralization because urban settings offer environments that are not completely removed from conditions a domestic species might encounter. For example, urban pigeons have spread across the world as humans first domesticated, then transported them for various uses [107], then repeatedly allowed them to feralize [108].

The abundance in cities of partially or fully domesticated taxa, often found at especially high densities because of their association with humans, creates a rich setting for feralization to occur (Figure 3). Some opportunities for feralization are intentional, as when urban domestic pigeons are usually allowed to fly free, even while they have a home coop. In other cases, human commensals such as several geckos in the genus *Hemidactylus,* have become so associated with human habitations that their common names include the word “house.” Even though they were never formally and intentionally domesticated, such species clearly coevolved with the human-created environments where they are now most often found. Of course, these animals are still found in non-anthropogenic habitats, where they can be locally abundant at times. Another unintentional impact is the bringing together of populations and even species that have not interacted in nature in many years, but are still capable of interbreeding, possibly producing even better-fit hybrids in their new environment. This seems to be the case with pythons now roaming the Everglades and nearby neighborhoods of Florida, USA [109].

In other cases, carelessness or thoughtlessness underlies the escape or release of domesticated individuals into the urban matrix. Many will not survive, others may remain localized, but some will disperse, often to other urban settings and sometimes to become damaging invasives [104]. A good example is the uncontrolled spreading of heavenly bamboo (*Nandina domestica*) in Florida [110] and elsewhere. An invasive ornamental, the plant’s berries are eaten by birds—many of them themselves non-natives—who disperse the seeds. Other releases are intentional, whether the result of religious sentiment, misguided animal welfare considerations, or other motivations. Traditional Buddhist wildlife release, for example, is a globally occurring phenomenon that can foster biological invasions [111], often in urban settings. In all these cases, the selection of an organism undergoes pre-release, to better fit it to intentional human desires or prevailing human conditions, thereby better preparing domesticated forms to better survive as feral organisms found in human vicinities [13,112] rather than being maladaptive [106]. 

From their urban strongholds, feralized species can further disperse into the countryside, potentially becoming damaging invasives, and this is the concern most commonly mentioned when conservation impacts of urban feralization are considered [113,114,115]. As urban areas are often perceived as having little conservation value anyway [86], less concern is evinced about the impact of feralized and invasive species on urban natives. Nonetheless, urban environments do hold many native species, a minority of them considered of high conservation concern, and the impacts of feral species on them are rarely discussed. For example, two management chapters in Adams et al. (2006, pp. 239–286 and 287–303) include some species identified as feral or meeting our definition [116]. And yet the case studies for cats, pigs, and Canada geese speculatively mention, but do not document, any actual impacts on urban non-humans. Similarly, of 11 mentions of feral pigeons in Murgui and Hedblom [117], not one details impacts on native species. 

The observation that we know relatively little about many urban issues in general, and those involving feral taxa in particular, is not new. Shochat et al. [118] suggested that competition from urban invasives, including feral species, may be an underlying cause of native species declines and concluded that, “*future research should give more attention to interspecific interactions in urban settings.*” There is some support for this hypothesis [119], though the authors “*argue that, in general, the role played by interspecific competition in current [urban] communities remains poorly understood.*” We believe there is considerable need for further studies on the impacts of feral species in general, but especially so in and related to urban settings. One exception, following the COVID-19 pandemic, is that considerably more attention has been paid to the ability of wildlife to carry diseases. The focus, however, has almost always been zoonotic diseases, ones that are carried by species in rural habitats and can affect humans once the two species come in contact [120]. Yet feral animals, particularly ones already found near humans in urban settings, have the potential to be vectors or carriers for diseases that also impact wildlife. For example, feral cats and dogs can serve as a reservoir for Chagas disease (*Trypanosoma cruzi*), a disease that can hurt people and also affects multiple species of wildlife [121]. Similarly, feral cats can transmit *Toxoplasma gondii* to both native birds and mammals, such as the endangered Hawaiian monk seal, *Neomonachus schauinslandi* [122], to free-ranging caracals (*Caracal caracal*) in South Africa [123], and to humans. The ability of feral taxa, especially non-mammalian ones, to transmit diseases to other species is another area in which we see a great need for additional research. This issue is of particular concern, we believe, in urban areas where feral species tend to be more densely populated.

## 7. Conclusions

Urbanization creates an ideal environment in which domesticated forms can become feralized and from which they can then spread and become damaging invasives. In reviewing the urban ecology literature, McDonald et al. [90] identified two major research gaps. First, “there is a need for more research in middle- and low-income countries on the impacts of urban growth on biodiversity.” Second, they saw a need for “more studies of indirect impacts of urban areas on biodiversity.” The focus of McDonald et al. [90] was the negative impacts of urbanization on wildlife in areas undergoing land-use conversion, but we see just as much application for them in the current context, where the literature is even sparser. We also agree with Yung et al. [124] that novel ecological systems tend to be poorly appreciated by the public at large. In fact, urban ecosystems are underappreciated by the scientific community as well [125]. Thus, the call of Yung et al. [124] for better stakeholder engagement, though not focused on urban ecosystems, certainly applies to them as well. As Yung et al. [124] point out, a city dweller is never far away from a UGS, which provides various human and wildlife benefits [126] and can act as a modest model for nature. We argue that this realization should also be extended to increasing appreciation within the scientific community. Feral taxa integrate well into a human-made environment, including novel ecological settings. We believe that such taxa could be of particular educational and other importance in the ongoing biodiversity crisis. There is a concern that humans might lose the perception that our efforts and resources as conservationists should address wild and native populations, instead of feral populations. This concern, however, needs to be addressed by managers and policymakers in future applications.

## Figures and Tables

**Figure 1 animals-13-00747-f001:**
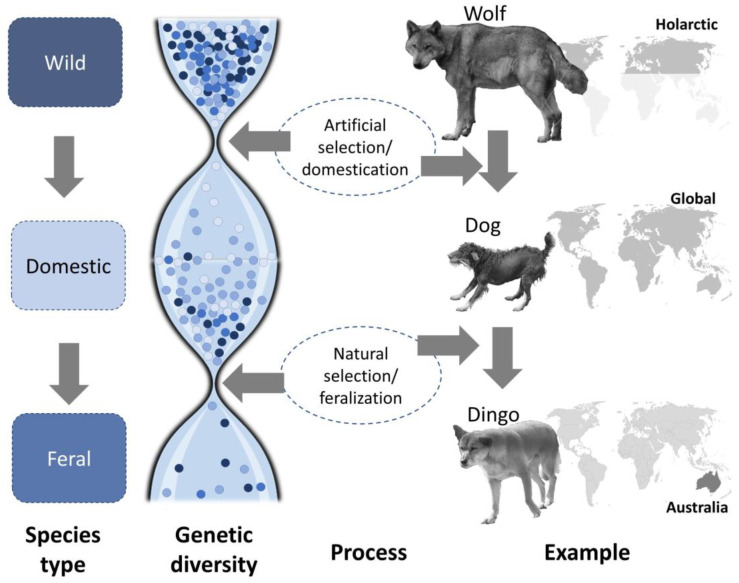
Conceptual framework using the wolf-dog-dingo continuum to illustrate the general processes of domestication (**top**) and feralization (**bottom**). Photos: T. Göttert.

**Figure 2 animals-13-00747-f002:**
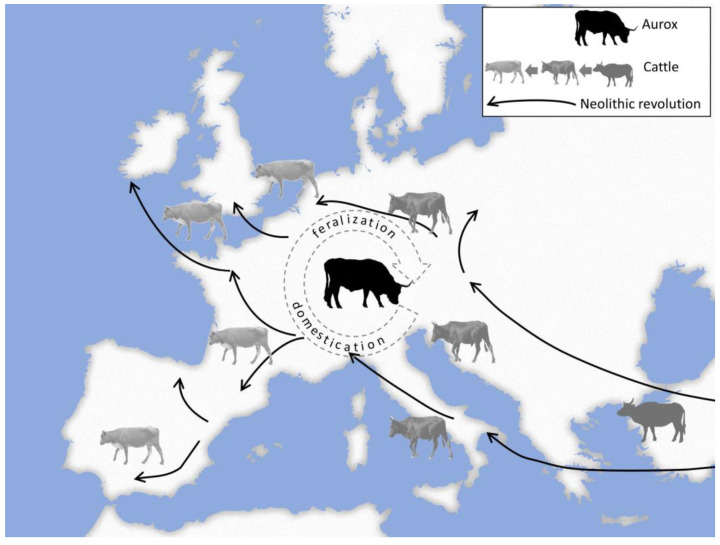
Ongoing interrelations between domestication and feralization as the Neolithic Revolution progressed into central Europe. Both processes were entangled as the wild taxon was around and the domestic taxon was not really detached from the natural environment. Domestic animals feralize and the hybrids of feralized domestic and wild animals may be re-incorporated into the human sphere as animal herders might consider this desirable. Photos: T. Göttert.

**Figure 3 animals-13-00747-f003:**
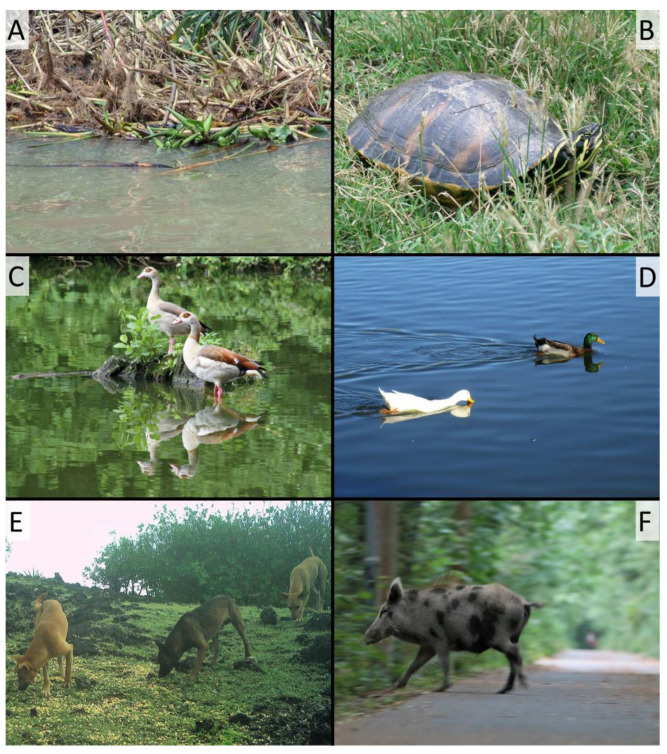
Selected examples of feral organisms from different parts of the world. (**A**): water hyacinth (*Eichhornia crassipes*) floating around at Lake Tana, Ethiopia; photo: G. Perry; (**B**): Florida red-bellied cooter (*Pseudemys nelsoni*), one of several found in the Road Town botanical garden pond, Tortola, British Virgin Islands; photo: G Perry; (**C**): Egyptian geese (*Alopochen aegyptiacus*) found at a pond in Saxony Anhalt, central Germany; photo: T. Göttert; (**D**): feral domestic ducks (*Anas platyrhynchos*) found at a water body in Lubbock, Texas, USA; photo: G. Perry; (**E**): feral dogs (*Canis lupus* f. familiaris) photographed at a bait station near Yigo, Guam; photo: USDA-Wildlife Services Guam; (**F**): subadult wild boar (*Sus scrofa*) photographed in the city of Berlin, Germany and showing the characteristic coat color pattern of domestic pigs; photo: T. Göttert.

**Figure 4 animals-13-00747-f004:**
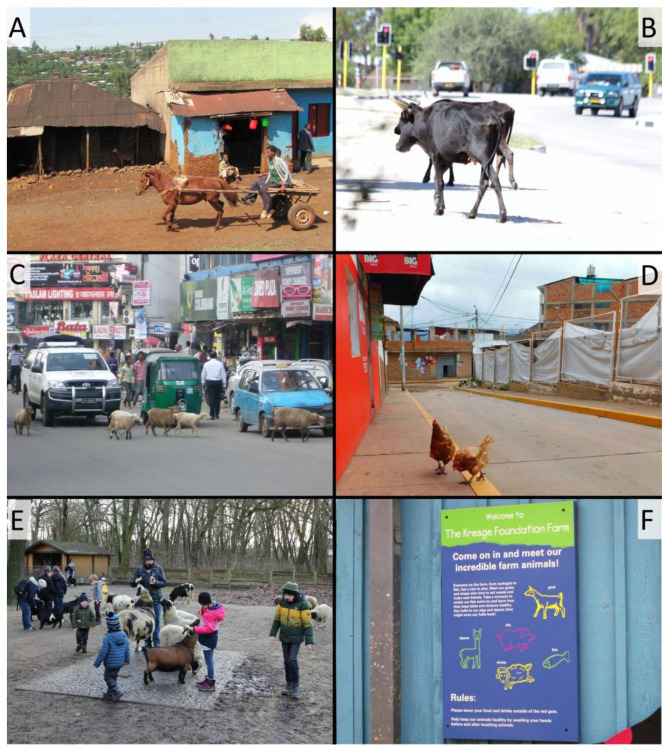
Livestock species found in urban areas in the Global South (**A**–**D**) and in the Global North (**E**,**F**). (**A**): horse pulling a wagon in Oromia region, Ethiopia; (**B**): cattle in Oshakati, Namibia; (**C**): sheep on the streets of Dhaka, Bangladesh; (**D**): chickens roaming in Cusco, Peru; (**E**): children’s zoo at Tierpark Berlin-Friedrichsfelde, Germany; (**F**): sign at children`s zoo at Dallas Zoo, USA; photos: (**A**,**E**): G. Perry; (**B**,**E**): T. Göttert; (**C**): C. Hobelsberger; (**D**): J.S. Rojas.

## Data Availability

Not applicable.

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
