# Peer review of "Going Wild in the City—Animal Feralization and Its Impacts on Biodiversity in Urban Environments"

_animals, 2023, doi:10.3390/ani13040747_

Round 1

Reviewer 1 Report

Excellent review on animal feralization in urban environments that worth be published in Animals. Below minor comments on some topics; these just pretend to depict my rather opposite opinion on some issues of the manuscript, but I acknowledge that they do not pretend diminish the quality of the paper or their possibilities to be published.

As a general comment, I recommend reducing the examples that address feralization in plants, and focus, even more, on animal species.

Line 16: “… but at other times, supporting ecosystem function or improving human well-being in urban settings”. I don’t like this point of view. There are currently many people that are increasingly perceiving feral populations, even domestic animal populations, as cases of biodiversity, either in urban or natural environments. I am concerned about that perception would be at the expense of natural biodiversity, the biodiversity afforded by wild (and native) population of organisms, and that would represent a risk. I agree that the presence of populations of several feral species within urban environments would increase, by an aesthetic mechanism, the human well-being in such environments. However, the presence of populations of the same number of species, but native and wild, at the same settings, would improve human well-being and natural biodiversity; I think conservationists must seek for the latter.

L. 28-29. From my point of view, the content of this sentence brings the same problem than the previous sentence. I can hardly admit that feral populations hold conservation value; and I don’t like “to rethink interrelations between humans and biodiversity”. We risk that many humans (and policy makers), would loss the perception than our efforts and resources as conservationists must address wild and native populations, instead of feral populations.

L. 125. I do not agree with the opinion by Price in his 1984’ paper. Domestic Japanese quails that became feral on Hawaiian islands can not be treated as wild Japanese quails, even after many generations. These quails have probably passed by a genetic bottleneck during the domestication process and be now genetically depressed. Moreover, they are introduced organisms in the Hawaiian islands, would compete with native organisms, change the food web, ….

L. 206. That would have an educational component. If “most people” perceive wild sheep in Western Europe as natural fauna elements, perhaps we must educate that people and inform them that these populations were introduced by man. Fortunately, conservationists and policy makers perfectly know that the Western European populations of that species were introduced.

L. 245. In dealing with the impact of the parakeets in Western Europe, please, add competition for nesting places and for food with native urban birds.

L. 373. This sentence is misleading. Perhaps these geckos are now most often found in artificial, human-made habitat, like urban habitat, for most people. But according to my experience with these lizards (of the genera Hemidactylus and Tarentola), they are more abundant and widespread in natural habitats, which is of more interest for the conservation of the species. The real abundance of a species in natural habitats is of more interest than the number of specimens found in a very particular, man-made, environment, under the point of view of the species conservation.

Hoping to have been useful.

Reviewer 2 Report

Overall:  Excellent paper providing a thorough review of some of the literature and perspectives in this field. 

Abstract: Very well written with a clear and concise summary of the topics covered in the paper and the relevance to the future of humankind and biodiversity. 

Introduction: 

Lines 41-47 – Add n a reference for the domestication definition and criteria. 

Domestication and Feralization:

Very well written.  The examples and explanations of feralization are easy to understand, even for the lay reader.  The authors also do a great job of explaining why feralization does not result in the original wild species.  While this could go into more detail, it is not needed for this paper and the topic was covered in enough detail for this particular topic. 

Conservation impacts of feralization:

This section does a great job of identifying the challenges of feralization and problems for native taxa in different environments.  This s an important issue when considering threats to more native species and impacts on biodiversity and conservation.

Urbanization:

Again, the authors do a great job of describing and defining urban environments and setting up the next few sections that tie in the meaning of feralization and the urban setting. 

Conservation impacts of urbanization:

This section builds on the last and details some of the species that have thrived in urban environments (or at least adjusted well).  The authors discuss some of the issues related to feral species in these dense human settings as well as some of the positive sides which helps balance out the discussion.

Domestication, feralization and urbanization:

The authors do a good job of clarifying the urban environment and the persistence of human influence and input.  This is a key component when discussion feral animals in this setting which would be different to discussion feral animals occupying abandoned urban environments. 
